# OpenReview forum: "AgentBench: Evaluating LLMs as Agents"
_ICLR.cc/2024/Conference — ICLR 2024 poster_

### Official Review · Reviewer_39pk · 2023-10-29

**Soundness:** 2 fair
**Presentation:** 1 poor
**Contribution:** 2 fair
**Rating:** 6
**Confidence:** 4

**Summary:**

The paper proposes a novel benchmark for evaluating the effectiveness of a given LLM to act as an agent in a specified environment. In particular, the current benchmark consists of eight tasks belonging to three main categories, namely code generation, game playing, and web browsing. Generally, each environment corresponds to a sequential decision-making problem, and, in some cases, the problems are best represented as POMDPs. The paper then provides a pretty exhaustive evaluation of many state-of-the-art LLMs. The results provide some interesting insights, including showing the status of the applicability of current state-of-the-art LLMs as agents and tasks where GPT-4 lags behind its predecessor, GPT-3.5-turbo.

**Strengths:**

The obvious strengths of the paper are its timeliness and significance. The use of LLMs as agents is a hot topic, and there is a need for benchmarking methods. The authors have put together a reasonably diverse and practical set of tasks and have put thought into creating a scalable and useful benchmarking system that new LLM-based agents can easily use. I would also like to praise the authors on the current set of evaluations, which is pretty extensive, and the authors have covered a lot of publicly available LLMs.

**Weaknesses:**

Now, coming to my concerns about the paper, my current worries can be broadly categorized into the following groups

Problem Selection and Quantification: The first issue is with the selection of task domains. The main motivating factor for the authors behind their current selection seems to be potential example scenarios where such LLM-based agents are currently being considered for deployment. While this can be a helpful metric, this approach overlooks an opportunity to curate datasets or even create random problem generators where you can accurately tune the complexity of the tasks being considered (quantified in objective terms such as computational complexity). This could have been done over the current task set (for example, one could quantify the complexity of certain OS operations by looking at factors like the size of filesystems, etc.), but it is currently missing. The current method of scoring each task depends on the abilities of the current set of LLMs being considered and isn’t an objective measure. One potential source for finding task sets where you can objectively quantify task hardness may be considering classical planning and logical reasoning literature. There, you can find many benchmark problems with varying degrees of complexity.

Dataset Collection: I was a bit surprised to find out that many datasets used in the benchmark were prepared using large language models, including GPT-4 and GPT-3.5. Wouldn’t this potentially influence or bias the results? For example, in the database task, data augmentation was carried out using GPT-3.5-turbo. This was also one of the tasks where GPT-3.5-turbo outperformed GPT-4. Is there any possibility that this is correlated?

Evaluation and Feedback Generation: In the two previous points, I have already pointed out some concerns with evaluation results. Now, I would like to bring up some other points related to evaluation. For one, I don’t know how fair it is to make a blanket claim that, currently, open-source LLMs fall behind closed-source ones when, by the authors' own admission, they only considered OSS models with less than or equal to 70B parameters. I would strongly encourage the authors to rephrase the claim. The evaluation and the benchmark also don’t currently appear to allow any form of task-specific fine-tuning. I didn’t see any discussion with respect to that topic. I also found the authors' choice to do prompt omission a bit surprising. While in the most general POMDP case, the full history of actions and observations is important for fully observable cases, can’t you create a new prompt where the current state of the task (along with any task specification) is fully summarized?

Clarity: Finally, the writing of the paper requires a lot of polish. There are numerous typos and malformed sentences littered throughout the paper, and the paper could benefit from thorough proofreading. However, the bigger concern I have is how the authors use certain terms and whether they imply technical meaning to them. There are many examples throughout, but let me point a few instances out. At one point during the analysis, the authors refer to incompleteness. Incompleteness is a technical term used in many areas of AI and computer science, including proof systems and sequential decision-making and planning. Are the authors referring to it in that technical sense? But then the incompleteness is related to the task, which doesn’t make sense from the traditional use of the term. Similarly, the term ‘turn-taking’ is usually associated with multi-agent games. While this kind of makes sense in the context of chat-based models, the authors use this term pretty liberally, which makes me think the authors may have instead meant to use it in the sense of multiple-step or sequential decision-making problems.

**Questions:**

I would ask the author to please respond to each question raised in the weakness section. I am particularly interested in the authors' thoughts about using LLMs in dataset generation.

---

> ### Author Response · Authors · 2023-11-19
> **Response to Reviewer 39pk (1/4)**
>
> Thank you for your thoughtful and constructive feedback and review! We really learn a lot from your comments, and spare no efforts during this response stage to make improvements according to your questions. Here are our responses:
>
> > Weakness 1:  **Problem Selection and Quantification.** The first issue is with the selection of task domains. The main motivating factor for the authors behind their current selection seems to be potential example scenarios where such LLM-based agents are currently being considered for deployment. While this can be a helpful metric, **this approach overlooks an opportunity to curate datasets or even create random problem generators where you can accurately tune the complexity of the tasks being considered (quantified in objective terms such as computational complexity)**. This could have been done over the current task set (for example, one could **quantify the complexity** of certain OS operations by looking at factors like the size of filesystems, etc.), but it is currently missing. The current method of scoring each task depends on the abilities of the current set of LLMs being considered and **isn’t an objective measure.** One potential source for finding task sets where you can objectively quantify task hardness may be ***considering classical planning and logical reasoning literature. There, you can find many benchmark problems with varying degrees of complexity.***
> >
>
> Thank you for your insightful feedback regarding the selection of task domains and the quantification of task complexity.
>
> 1. **Choice of Scenarios**:
>     - We agree that it is not feasible to cover every potential scenario in our benchmark. However, we carefully selected well-known, real-world scenarios after research. Our decision-making process is grounded in two fundamental principles:
>     - Recognition Criterion: Priority is given to task types universally recognized as pivotal within the language agent community. For instance, benchmarks like NL2Bash[4] and InterCode[5] emphasize the significance of Operating System (OS) tasks. In a similar vein, Wob[6], MiniWob++[7], Webshop[8], Mind2Web[9], and WebArena[10] draw attention to the vital nature of web-based tasks.
>     - LLM Applicability: A crucial factor in our selection process is the suitability of tasks for the existing capabilities of Large Language Models (LLMs), specifically in terms of their ability to handle few-shot text prompts in multi-round interactions. For example, while datasets like Minedojo[5] offer more complex challenges in open-world game environments, they often necessitate visual information processing, which is beyond the scope of current LLM capabilities.
> 2. **Difficulty Quantification:**
>
>     Thanks for your great suggestion. We thoroughly re-examine our task settings to make the following improvements:
>
>     - **Tasks that are objectively controllable:** For many tasks, we can give an objective definition of difficulty in a specific domain. For example, for an MDP with known environment state transition and limited action space, after limiting interaction rounds, we can calculate the expected score in the same way as the score of Curated List in Text World[1] is defined, and normalize the real score on this basis. .
>     - **Tasks that reflect complex real-world needs**: As our aim is to create diverse tasks across multiple domains, we find it infeasible to uniform difficulty quantification for all tasks we consider. For example, tasks in OS operations can span various domains like file manipulation, user management, and network settings, each with inherently different complexity metrics. These tasks cannot be properly assigned difficulty control, or their comprehensiveness to reflect real-world cases would be significantly harmed.

---

> ### Author Response · Authors · 2023-11-19
> **Response to Reviewer 39pk (2/4)**
>
> 3. **Task Selection and Random Generation**:
>     - Your recommendation of random task generation is valuable. Many tasks in our study could indeed be generated randomly by setting certain parameters:
>         - OS: Introduce randomness in file operations and design task templates with variable slots.
>         - Database: Similar to OS.
>         - Digital Card Game: Vary team compositions, parameters (e.g., health, attack), and opponent strategies.
>     - For the sake of evaluation fairness and reproducibility, we pre-fixed some datasets or presets to ensure uniform evaluation across different models.
>     - To enhance the utility in creating diverse problems with specific configuration, we are introducing configurable options in our task settings. These options enable the generation of tasks based on specific parameters such as difficulty. For example, in Digital Card Game, we have updated the concept of difficulty and the method for generating random games based on difficulty. Here's a more detailed explanation:
>         - The difficulty of a combat is determined by comparing the Combat Powers of the competing teams. This measure is based on the expected survival rounds of the teams. Specifically, the difficulty faced by a friendly team \( F \) against a hostile team \( H \) is expressed as:
>             - $\text{Difficulty}(H \text{ against } F) = \frac{\text{Power}(H)}{\text{Power}(F)}$
>             - Here, $\text{Power}(T)$ is calculated as the product of total Health Points (HP) and average Attack Points (ATK) for each character $c$ in team $T$:
>                 - $\text{Power}(T) = \frac{1}{\\#T} \sum_{c\in T}HP_c \sum_{c\in T}ATK_c$
>         - For a comprehensive understanding and further technical details, please refer to Appendix E.4.
>
> Thank you once again for your constructive feedback.
>
> > Weakness 2:  Dataset Collection: I was a bit surprised to find out that many datasets used in the benchmark were **prepared using large language models**, including GPT-4 and GPT-3.5. **Wouldn’t this potentially influence or bias the results?** For example, in the database task, data augmentation was carried out using GPT-3.5-turbo. This was also one of the tasks where GPT-3.5-turbo outperformed GPT-4. Is there any possibility that this is correlated?
> Question:  I am particularly interested in the authors' thoughts about using LLMs in dataset generation.
> >
>
> We have also taken into consideration the potential biases that may arise from using Large Language Models (LLMs) for data augmentation, as you mentioned. During the data generation process, we have paid special attention to this issue. Taking the data generation process of DB as an example, the augmentation can be divided into two steps: table augmentation and row labeling.
>
> - During the table augmentation phase, we ask LLMs to generate similar data entries based on existing rows. After generation, we verify the legitimacy of these data and incorporate them into the table. The purpose of this step is to increase the complexity of the task. Since the LLM simply imitates and repeats based on existing entries, this process should not introduce bias.
> - In the INSERT and UPDATE tasks of DB, we used LLMs to generate questions and answers. Specifically, we provide LLMs with header information and row data of the table. We first ask the LLM to generate an SQL statement. If the statement is valid and changes the database, we retain this data. On this basis, we further require the LLM to generate an Instruction describing the changes made in the SQL.
>
> In summary, during the data augmentation phase, we provide as much information as possible to the LLM, enabling it to perform a relatively simple process of generating from a larger to a smaller amount of information. In contrast, during the evaluation phase, we provide limited information to the LLM, requiring it to infer from a smaller to a larger amount of information, a more complex and challenging process. This approach helps to minimize potential biases introduced during the data augmentation stage.
>
> To add credibility, we re-executed a small batch of data labeling for INSERT and UPDATE in DB using claude-2 and re-tested on gpt-4 and gpt-3.5-turbo. The results showed that gpt-3.5-turbo still outperforms gpt-4 in the UPDATE category, as it did originally. This indicates that the superiority of gpt-3.5-turbo over gpt-4 is not due to biases introduced in the data augmentation process, but rather due to other reasons. (Cf. Appendix C.4)

---

> > ### Author Response · Authors · 2023-11-19
> > **Response to Reviewer 39pk (3/4)**
> >
> > > Weakness 3.1:  Evaluation and Feedback Generation: In the two previous points, I have already pointed out some concerns with evaluation results. Now, I would like to bring up some other points related to evaluation. For one, I don’t know how fair it is to make a blanket claim that, currently, open-source LLMs fall behind closed-source ones when, by the authors' own admission, **they only considered OSS models with less than or equal to 70B parameters.** I would strongly encourage the authors to rephrase the claim. **The evaluation and the benchmark also don’t currently appear to allow any form of task-specific fine-tuning**. I didn’t see any discussion with respect to that topic.
> > >
> >
> > Thank you for pointing this out. However, in our defence, in fact, the size smaller than 70 billion has already encompassed the majority of open-source LLMs that have undergone fine-tuning for chat or instruction tasks, which constitutes the focus of our testing. Nevertheless, for the sake of rigor, we have revised all the statements in the paper regarding this conclusion to employ more cautious and circumspect expressions.
> >
> > Regarding the issue of task-specific fine-tuning that you mentioned, it is worth noting that the trajectory formed by our benchmark is indeed applicable for fine-tuning. Some recent efforts have also explored attempts to fine-tune on the trajectory of the agent, such as FireAct[2] and AgentTuning[3]. Due to space constraints and the limited extent of our exploration in this regard, we refrain from making unwarranted claims and, therefore, have not extensively discussed the possibilities in this aspect in the main body of the paper.
> >
> > > Weakness 3.2:  I also found the authors' choice to do prompt omission a bit surprising. While in the most general POMDP case, **the full history of actions and observations is important for fully observable cases**, can’t you **create a new prompt where the current state of the task (along with any task specification) is fully summarized**?
> > >
> >
> > Thank you for your insightful comment on prompt omission. To clarify, the majority of tasks designed in our study are intended to be completed within 2000 tokens. In fact, our analysis of the interaction trajectories supports this design choice. For completed tasks, we observed an average of 7.95 rounds per task and an average token usage of 2220.1. Additionally, it was rare for the model to use more than 3000 tokens. These findings are detailed in Appendix J.2.3 of our revised version.
> >
> > Therefore, if the total token count in an interaction exceeds 3500, it indicates a low likelihood of the task being completed. This approach to omission minimally impacts our experimental results. Moreover, this strategy was consistently applied across all models, ensuring a fair evaluation framework.
> >
> > Furthermore, another interesting observation is made, as detailed in Appendix J.2.4 of our revised version. We find that the majority of instances where models failed to complete tasks were due to a tendency to repeat content already generated. This insight further supports our approach, as a summary that condenses only the current state of the task would not necessarily mitigate this issue of content repetition.
> >
> > We believe these explanations address your concerns, and we thank you again for your constructive feedback, which has helped improve our analysis and manuscript.

---

> > > ### Author Response · Authors · 2023-11-19
> > > **Response to Reviewer 39pk (4/4)**
> > >
> > > > Weakness 4:  Clarity. Finally, the writing of the paper requires a lot of polish. There are **numerous typos** and **malformed sentences** littered throughout the paper, and the paper could benefit from thorough proofreading. However, the bigger concern I have is how the authors **use certain terms and whether they imply technical meaning to them**. There are many examples throughout, but let me point a few instances out. At one point during the analysis, the authors refer to incompleteness. Incompleteness is a technical term used in many areas of AI and computer science, including proof systems and sequential decision-making and planning. Are the authors referring to it in that technical sense? But then the incompleteness is related to the task, which doesn’t make sense from the traditional use of the term. Similarly, the term ‘turn-taking’ is usually associated with multi-agent games. While this kind of makes sense in the context of chat-based models, the authors use this term pretty liberally, which makes me think the authors may have instead meant to use it in the sense of multiple-step or sequential decision-making problems.
> > > >
> > >
> > > Thank you for your careful and detailed reading! We have revisited and polished the expression in our paper, rectifying any grammatical errors. Regarding the term "turn-taking" that you mentioned, in our paper, "turn" primarily refers to a round of interaction between the user (in the context of this benchmark, it refers to the task environment) and the chat model, rather than referring to the turn of each agent in a multi-agent interaction. To ensure clarity in terminology throughout the text, we have replaced the term "turn" with "round" in all instances where it refers to a single interaction with the chat model. We hope this enhances the reading experience for our audience.
> > >
> > > ## General
> > >
> > > We are deeply grateful for your meticulous and thorough review of our work. Your insightful suggestions have been immensely beneficial, significantly enhancing the completeness and perfection of our efforts. We hope that our responses have satisfactorily addressed your concerns. We cannot thank you enough if you could raise your score to support us.
> > >
> > > ## References
> > >
> > > [1] Côté, Marc-Alexandre, et al. "Textworld: A learning environment for text-based games." *Computer Games: 7th Workshop, CGW 2018, Held in Conjunction with the 27th International Conference on Artificial Intelligence, IJCAI 2018, Stockholm, Sweden, July 13, 2018, Revised Selected Papers 7*. Springer International Publishing, 2019.
> > >
> > > [2] Chen, Baian, et al. "FireAct: Toward Language Agent Fine-tuning." *arXiv preprint arXiv:2310.05915* (2023).
> > >
> > > [3] "AgentTuning: Enabling Generalized Agent Abilities for LLMs." *arXiv preprint arXiv:2310.12823* (2023).
> > >
> > > [4] Lin, Xi Victoria, et al. "NL2Bash: A Corpus and Semantic Parser for Natural Language Interface to the Linux Operating System." *Proceedings of the Eleventh International Conference on Language Resources and Evaluation (LREC 2018)*. 2018.
> > >
> > > [5] Yang, John, et al. "InterCode: Standardizing and Benchmarking Interactive Coding with Execution Feedback." *arXiv preprint arXiv:2306.14898* (2023).
> > >
> > > [6] Shi, Tianlin Tim, et al. "World of bits: an open-domain platform for web-based agents." *Proceedings of the 34th International Conference on Machine Learning-Volume 70*. 2017.
> > >
> > > [7] Liu, Evan Zheran, et al. "Reinforcement Learning on Web Interfaces using Workflow-Guided Exploration." *International Conference on Learning Representations*. 2018.
> > >
> > > [8] Yao, Shunyu, et al. "Webshop: Towards scalable real-world web interaction with grounded language agents." *Advances in Neural Information Processing Systems* 35 (2022): 20744-20757.
> > >
> > > [9] Deng, Xiang, et al. "Mind2Web: Towards a Generalist Agent for the Web." *arXiv preprint arXiv:2306.06070* (2023).
> > >
> > > [10] Zhou, Shuyan, et al. "WebArena: A Realistic Web Environment for Building Autonomous Agents." *NeurIPS 2023 Foundation Models for Decision Making Workshop*. 2023.

---

> ### Author Response · Authors · 2023-11-22
> **Thanks for your great review!**
>
> Dear Reviewer,
>
> Thank you very much again for your kind and valuable reviews! We hope our previous response addresses your questions, from which this work has benefited a lot. If possible, we would be deeply grateful if you could help share suggestions to further make AgentBench a stronger work!
>
> Thank you again for your encouraging comments!
>
> Authors of Submission7101

---

> > ### Comment · Reviewer_39pk · 2023-11-22
> > **Reply**
> >
> > I think the authors for their reply; I am increasing the score a bit, especially because of the new results related to the DB update. In general, I still think the paper would greatly benefit from a thorough revision.

---

### Official Review · Reviewer_WbYN · 2023-10-29

**Soundness:** 4 excellent
**Presentation:** 4 excellent
**Contribution:** 4 excellent
**Rating:** 8
**Confidence:** 4

**Summary:**

This paper presents AgentBench, a multi-dimensional evolving benchmark that currently consists of 8 distinct environments to assess LLM-as-Agent's reasoning and decision-making abilities in a multi-turn open-ended generation setting. Extensive evaluations over 27 API-based and open-sourced (OSS) LLMs are token by the authors. The authors found that 1) while top commercial LLMs present a strong ability of acting as agents in complex environments, there is a significant disparity in performance between them and OSS competitors; 2) poor long-term reasoning, decision-making, and instruction following abilities are the main obstacles for developing usable LLM agents; 3) Training on code and high quality multi-turn alignment data could improve agent performance. Improtantly, the datasets, environments, and an integrated evaluation package for AgentBench are released.

**Strengths:**

1. AgentBench is a multi-dimensional evolving benchmark (Including 3 types of and totally 8 environments).
2. The authors do a lot of evaluations (27 LLMs on all environments), and demonstrate many useful insights for LLM-based Agents.
3. The authors kindly give a weight for each environment for a fair comparsion.
4. The authors releas the code and datasets.

**Weaknesses:**

I like this work very much, from presentation to the solid work.

If I must point out some weaknesses, I would like to encourage the authors to add more related works about the DB tasks (since I found that there are many recent works are not mentioned). For example, the three papers mentioned in Section 3.1 are from 2017/2017/2021, but the are many recent works study SQL generation, e.g. DIN-SQL/DIAL-SQL/TPTU [1,2,3].  Besides, the authors mentioned that "However, few previous code evaluation frameworks consider multi-turn interactions", but as far as I know, DIAL-SQL and TPTU are also multi-turn interactions with sql database (possibly with error feedback).

[1] DIN-SQL: Decomposed In-Context Learning of Text-to-SQL with Self-Correction
[2] DIAL-SQL: Text-to-SQL Empowered by Large Language Models: A Benchmark Evaluation
[3] TPTU: Task Planning and Tool Usage of Large Language Model-based AI Agents

**Questions:**

The is no question

---

> ### Author Response · Authors · 2023-11-19
> **Response to Reviewer WbYN**
>
> Thanks for your appreciation of our work and valuable suggestions! Here are our responses:
>
> > Weakness 1:  If I must point out some weaknesses, I would like to encourage the authors to **add more related works about the DB tasks** (since I found that there are many recent works are not mentioned). For example, the three papers mentioned in Section 3.1 are from 2017/2017/2021, but the are many recent works study SQL generation, e.g. DIN-SQL/DIAL-SQL/TPTU [1,2,3].
> >
> >
> > [1] DIN-SQL: Decomposed In-Context Learning of Text-to-SQL with Self-Correction
> > [2] DIAL-SQL: Text-to-SQL Empowered by Large Language Models: A Benchmark Evaluation
> > [3] TPTU: Task Planning and Tool Usage of Large Language Model-based AI Agents
> >
>
> Really thank you for your valuable suggestions. In our revised version, We have updated section 3.1 and these recent works about DB are cited.
>
> > Weakness 2:  Besides, the authors mentioned that "**However, few previous code evaluation frameworks consider multi-turn interactions**", but as far as I know, DIAL-SQL and TPTU are also multi-turn interactions with sql database (possibly with error feedback).
> >
>
> Thank you for highlighting the works of Dial-SQL and TPTU. We appreciate your insights and acknowledge the relevance of these contributions in the field of code evaluation on databases. However, our research has a distinct focus, particularly on multi-turn interactions and code execution, which sets it apart from these studies:
>
> 1. **TPTU: Large Language Model-based AI Agents for Task Planning and Tool Usage**
>     - TPTU primarily investigates how Large Language Models (LLMs) can be leveraged for task planning and tool usage. Although it encompasses multi-turn interactions, the central aim is not explicitly focused on multi-turn code execution, a key element of our study.
> 2. **Text-to-SQL Empowered by Large Language Models: A Benchmark Evaluation**
>     - Our understanding is that the primary contribution of Dial-SQL is concerning the selection of examples in the domain of In-Context Learning. This is distinct from our research, which delves deeper into the complexities of multi-turn interactions in practical coding environments.
>
> We hope this clarification helps in understanding the unique aspects of our research. We are grateful for your input, which has helped us improve the clarity of our work.

---

> > ### Comment · Reviewer_WbYN · 2023-11-20
> > **Thanks for the clarification. I support the acceptance of this paper.**
> >
> > Thanks for the clarification provided by the authors. It does help in understanding the unique aspects of AgentBench.
> >
> > I have also read the reviews from other reveiwers. Although some of them are negative, I found this research is clearly helpful for benchmarking LLM-based agents. So I support the acceptance of this paper.

---

> ### Author Response · Authors · 2023-11-22
> **Thanks for your great review!**
>
> Dear Reviewer,
>
> Thank you very much again for your kind and valuable reviews! We hope our previous response addresses your questions, from which this work has benefited a lot. If possible, we would be deeply grateful if you could help share suggestions to further make AgentBench a stronger work!
>
> Thank you again for your encouraging comments!
>
> Authors of Submission7101

---

### Official Review · Reviewer_eDe1 · 2023-10-30

**Soundness:** 3 good
**Presentation:** 3 good
**Contribution:** 3 good
**Rating:** 6
**Confidence:** 4

**Summary:**

The authors propose a benchmark of 8 distinct environments to test LLMs' ability to operate various APIs.


# After Rebuttal

I appreciate the author's efforts at making the paper much better.

Overall, I think the work does have a contribution, despite the weaknesses I mentioned in the review still exist. I will retain my score for the paper.

**Strengths:**

1. This is quite a unique type of benchmark, and could have profound implications for future LLM research or LLM as agent applications.
2. The benchmark captures a variety of tasks.

**Weaknesses:**

1. The benchmark does not seem to offer any insights for improvement. (i.e. If my model is not doing well on web-browsing, what should I do?)

2. The embodied tasks seem quite contrived. AlfWorld drops all 2d/3d aspects of the environment and could be mastered by a fine-tuned GPT-2 [1].

3. The benchmark seems to be mostly coding based. Non-coding LLMs could potentially still behave as good agents, but would underperform on this benchmark.

Overall, I like the paper direction. All the below weaknesses should be considered `Minor Issues', but I feel strongly about these aspects and hope that the authors would address them.

1. The use of abbreviations. I find the abbreviations "OS DB KG DCG LTP HH WS WB" make little sense to me. It took me a long time to dig through different pages to understand the benchmark. I hope the authors would re-format the tables for better readability.

2. Figure (a) is really hard to read. I struggle to tell the colors apart. I suspect that this figure is not color-blind friendly.

[1] Language Models are Few-Shot Butlers. Vincent Micheli, François Fleuret

**Questions:**

1. Is there a high-score for these benchmarks? For example, human expert score.

Please focus on the Weaknesses and improve the presentation of the paper. I have no questions otherwise.

---

> ### Author Response · Authors · 2023-11-19
> **Response to Reviewer eDe1 (1/2)**
>
> We sincerely appreciate the points you've raised; they are of great value to our discussion and have offered important insights. Thank you for your insightful and well-articulated points. It's clear that a lot of thought and consideration has gone into your perspective.
>
> > Weakness 1:  The benchmark **does not seem to offer any insights for improvement.** (i.e. If my model is not doing well on web-browsing, what should I do?)
> >
>
> Thank you for you input. We address your concerns as follows:
>
> - **Observations from Interaction Trajectories**: Our research offers valuable insights, particularly in scenarios where tasks are not successfully completed. We've identified two primary issues:
>     - **Instruction Following**: Models often struggle with adhering to given instructions. This leads to outputs that are either format-incompatible or lack a valid action. (Cf. Appendix J.2.1)
>     - **Output Repetition**: There's a tendency for models to repeat previous outputs after a certain number of interactions. This repetition prevents task completion and causes interactions to reach the maximum turn limit. (Cf. Appendix J.2.3)
>     - **Code Training with Ambivalent Impacts**: We show that code training helps certain tasks (e.g., Database, WebShop). However, its introduction might harm other tasks (e.g., Digital Card Games, Operating System) that require more ability in interacting. Later work should be aware of this double-edged sword. (Cf. Section 4.3 & Appendix J.2.5)
>     - **High-Quality Alignment Data**: By comparing models like vicuna-13b and llama-2-13b (which share the same base model), we show that alignment data distilled from GPT-4 (e.g., ShareGPT), significantly enhances LLM performance. Vicuna-13b, aligned with ShareGPT data, notably outperforms its counterpart and rivals larger models. (Cf. Section 4.3)
> - **Implications for Fine-Tuning**: These observations suggest key areas for fine-tuning LLMs-as-Agents, one is enhancing instruction compliance, and another is addressing repetitive outputs.
> - **Future Work and Community Contribution**: While our current focus is on establishing a robust agent benchmark for the community, we acknowledge the importance of fine-tuning exploration. We plan to delve into this in our future work. It's also worth mentioning that recent studies like **FireAct[3]** and **AgentTuning[4]** have already been inspired and begun exploring fine-tuning approaches for LLMs-as-Agents.
>
> In summary, our work lays the groundwork for future advancements in fine-tuning LLMs-as-Agents, offering a clear direction for addressing key challenges identified in our research.
>
> > Weakness 2:  The **embodied tasks seem quite contrived. AlfWorld** drops all 2d/3d aspects of the environment and could be mastered by a fine-tuned GPT-2 [1].
> [1] Language Models are Few-Shot Butlers. Vincent Micheli, François Fleuret
> >
>
> Thank you for your further insights regarding the selection of ALFWorld as an evaluation task in our study. We appreciate the opportunity to clarify the reason of choosing it.
>
> While ALFWorld, as a text-based environment, does simplify the complexity, it is still a problem relevant and challenging for language models, particularly those not fine-tuned on the task. It requires strategic planning and decision-making within a given scenario. As illustrated in Appendix J.2.2, these tasks demand a level of cognitive engagement and planning ability that goes beyond the scope of basic language understanding, thereby offering a valuable framework for evaluating the depth and adaptability of language models.
>
> As a result, ALFWorld has been widely adopted in recent typical LLM-as-Agent study (e.g., ReAct[1], Reflextion[2]), and we thus adopt it in AgentBench in an effort to unify the evaluation and results for later LLMs. However, we do agree that it might not be so challenging for LLMs after task-specific fine-tuning, and we are closely working on finding more complement datasets for the task in our next planned update.

---

> > ### Author Response · Authors · 2023-11-19
> > **Response to Reviewer eDe1 (2/2)**
> >
> > > Weakness 3:  The benchmark seems to be **mostly coding based**. Non-coding LLMs could potentially still behave as good agents, but would underperform on this benchmark.
> > >
> >
> > Thanks for your valuable comment. Mindful of this important aspect, we wish to clarify that we have conscientiously designed AgentBench to be not mostly coding based, ensuring a balanced representation of tasks.
> >
> > In our benchmark, only 3 out of 8 tasks, specifically Operating system, Database, and Knowledge Graph, involve coding aspects. These tasks are designed to evaluate the agent ability of language models when they are grounded on coding. However, it is crucial to highlight that a significant portion (5 out of 8) of our benchmark, including Digital Card Game, Lateral Thinking , House Holding, Web Shopping, and Web Browsing, does not involve coding at all. These tasks focus on different aspects like decision-making, language comprehension, and reasoning, offering a comprehensive evaluation of language models in non-coding contexts.
> >
> > This balance between coding and non-coding tasks ensures a holistic assessment of language models, capturing their versatility and range of capabilities. We believe this approach accurately reflects the diverse applications and potential of contemporary language models.
> >
> > > Issue 1:  The use of abbreviations. I find the **abbreviations** "OS DB KG DCG LTP HH WS WB" make little sense to me. It took me a long time to dig through different pages to understand the benchmark. I hope the authors would re-format the tables for better **readability**.
> > >
> >
> > We sincerely apologize for any misunderstanding and confusion caused by the abbreviations. We have replaced all abbreviations with their full forms in the paper.
> >
> > > Issue 2:  Figure (a) is really hard to read. I struggle to **tell the colors apart**. I suspect that this figure is **not color-blind friendly**.
> > >
> >
> > We sincerely apologize. We have chosen a clearer and more distinct method to present this figure to ensure better readability and accessibility, including reduce 9 displayed models to 6, use black-and-white to indicate API-based LLMs, and use seaborn’s `colorblind` palette to color OSS LLMs.
> >
> > > Question 1:  Is there a high-score for these benchmarks? For example, **human expert score.**
> > >
> >
> > We highly value your concerns. We also considered setting up a human expert score, but after comprehensive consideration, we found that the complete dataset contains about a thousand evaluation entries. If done by humans, each entry would approximately take 5-20 minutes, amounting to around 200 hours to complete the entire benchmark. Even a smaller dev dataset would require about 50 hours. Additionally, some tasks in this dataset involve more complex knowledge (such as Ubuntu operations, SQL commands), thus requiring a high level of expertise. To collect a relatively reliable Human expert data (e.g., 385 people for 95% confidence level and 5% confidence interval), the overall cost, even by the lowest estimate, is prohibitive (385 people * 30h * $5/h = $57,750). Therefore, we did not include this data in the paper. However, we will still strive to gradually collect this data in the future.
> >
> > ## References
> >
> > [1] Yao, Shunyu, et al. "ReAct: Synergizing Reasoning and Acting in Language Models." *The Eleventh International Conference on Learning Representations*. 2022.
> >
> > [2] Shinn, Noah, et al. "Reflexion: Language agents with verbal reinforcement learning." *Thirty-seventh Conference on Neural Information Processing Systems*. 2023.
> >
> > [3] Chen, Baian, et al. "FireAct: Toward Language Agent Fine-tuning." *arXiv preprint arXiv:2310.05915* (2023).
> >
> > [4] "AgentTuning: Enabling Generalized Agent Abilities for LLMs." *arXiv preprint arXiv:2310.12823* (2023).

---

> ### Author Response · Authors · 2023-11-22
> **Thanks for your great review!**
>
> Dear Reviewer,
>
> Thank you very much again for your kind and valuable reviews! We hope our previous response addresses your questions, from which this work has benefited a lot. If possible, we would be deeply grateful if you could help share suggestions to further make AgentBench a stronger work!
>
> Thank you again for your encouraging comments!
>
> Authors of Submission7101

---

### Official Review · Reviewer_39aA · 2023-10-31

**Soundness:** 3 good
**Presentation:** 3 good
**Contribution:** 3 good
**Rating:** 8
**Confidence:** 3

**Summary:**

This paper proposes a new benchmark AgentBench to evaluate LLM as agents. The benchmark covers 8 environments in 3 categories, including some datasets curated or adapted from existing works. The authors benchmark 27 models and show that closed-source API-based LLMs are far better than open-sourced ones.

**Strengths:**

- The paper is well-written and easy to follow.
- The benchmark covers diverse tasks and includes a well-designed HTTP evaluation interface. Overall it seems well thought through.
- The experiment results over 27 models could be very useful reference for LLM development

**Weaknesses:**

- The benchmark seems to use the same prompt for all models, which might give an unfair advantage to the model where these prompts were developed for.
- There could be data leakage to the tasks selected from the pretraining data over the internet.

**Questions:**

- How is the success rate calculated with the runs failed to complete?
- Which model does table 4 correspond to?

---

> ### Author Response · Authors · 2023-11-19
> **Response to Reviewer 39aA**
>
> Thanks for your appreciation of our work and valuable suggestions! Here are our responses:
>
> > Weakness 1:  The benchmark seems to use the **same prompt for all models**, which might give an unfair advantage to the model where these prompts were developed for.
> >
>
> Thanks for your pointing out. In fact, our task prompts are chosen based on a combination of representative LLMs (rather than an individual series of models) to ensure their robustness. We experimented with various prompt paradigms, such as ReAct[1] and Chain of Thought (CoT)[2], and tested them across multiple models including claude-2, gpt-4, gpt-3.5-turbo, llama-2 and so on to ensure their universal validness. We did not select task prompts specifically tailored to any particular model. All models were tested under relatively equal conditions.
>
> > Weakness 2:  There could be **data leakage to the tasks** selected from the pretraining data over the internet.
> >
>
> Thanks for your pointing out. We acknowledge that there is a risk of data leakage when selecting tasks from internet-based pretraining datasets. However, a significant difference of AgentBench to other benchmarks is that to solve AgentBench requires leakage of groundtruth trajectories based on our dynamic environments (e.g., filenames and paths in linux dockers of our OS dataset), which does not exist on the web as text corpora.
>
> As a result, we believe for AgentBench, there is very minimal risks of data leakage in the evaluation.
>
> > Question1:  How is the **success rate** calculated with the runs failed to complete?
> >
>
> We calculate success rate as follows:
>
> $SR = \frac{C}{N}$,
>
> where $C$ represents the count of correct results, and $N$ denotes the total count of samples.
>
> More specifically, for tasks whose evaluation metrics are success rate, the following situations may occur after each sample has finished:
>
> - Completed:  the final status is a termination status (e.g., submit an answer in OS or DB)
>     - Correct:  the final status is accepted as a correct solution.
>     - Wrong:  the final status is NOT accepted as a correct solution.
> - Context Limit Exceeded
> - Invalid Format
> - Invalid Action
> - Task Limit Exceeded
>
> Only **Correct** results are the part of successful ones, meaning:
>
> $SR = \frac{\\#Correct}{\\#Completed + \\#CLE + \\#IF + \\#IA + \\#TLE}$
>
> > Question2:  Which model does **table 4** correspond to?
> >
>
> Thank you for pointing this out. Table 4 represents an overall ratio across all models. We have fixed this ambiguity in the paper.
>
> ## References
>
> [1] Yao, Shunyu, et al. "ReAct: Synergizing Reasoning and Acting in Language Models." *The Eleventh International Conference on Learning Representations*. 2022.
>
> [2] Wei, Jason, et al. "Chain-of-thought prompting elicits reasoning in large language models." *Advances in Neural Information Processing Systems* 35 (2022): 24824-24837.

---

> > ### Comment · Reviewer_39aA · 2023-11-22
> >
> > Thank you for the detailed response. I will maintain the current score.

---

> ### Author Response · Authors · 2023-11-22
> **Thanks for your great review!**
>
> Dear Reviewer,
>
> Thank you very much again for your kind and valuable reviews! We hope our previous response addresses your questions, from which this work has benefited a lot. If possible, we would be deeply grateful if you could help share suggestions to further make AgentBench a stronger work!
>
> Thank you again for your encouraging comments!
>
> Authors of Submission7101

---

### Official Review · Reviewer_hDNu · 2023-11-01

**Soundness:** 2 fair
**Presentation:** 2 fair
**Contribution:** 2 fair
**Rating:** 3
**Confidence:** 4

**Summary:**

This paper proposes a benchmark for LLM as Agents. The paper selects a variety of text-based benchmarks and evaluates a series of models on this setting.

**Strengths:**

- The paper proposes a comprehensive benchmark with evaluation results on a wide-set of tasks

**Weaknesses:**

- The contributions of the paper seem very limited -- the paper does not propose any new technical insights and simply applies a variety of LLMs on many existing environments.

- The analysis is not particularly insightful and I'm not sure if the conclusions are fully accurate from the analysis. For instance, task length exceeded may just be due to the fact that many LLMs are trained on short fixed context lengths. In setting such as this, it would be better to first summarize the long context to keep inputs in distribution to the input LLM.

- Similarily, the fact that code training helps models is relatively well known. Codellamma performing worse than llamma 2 may just be due to weakness in the original pretrained model that the model is fine-tuned on. Overall, the conclusions in the paper mostly just follow the general trend of the more capable the LLM, the better the performance on the benchmark

- In the related work section, it would be good to also add some references to multiagent approaches to reasoning such as multiagent debate. Also age 2023 seems to be incorrectly formulated.

- Some of the references in the paper are incorrectly cited. For instance, in the intro, Reed et al. 2022 does not use a LLM to learn an embodied policy, a better reference would be [1]. Similarily, Ahn et al 2022 does not use a complex multi-modal simulator based off games but rather a real robot.

[1] Pre-Trained Language Models for Interactive Decision-Making. NeurIPS 2022

**Questions:**

1) Have you tried seeing the results of each method assuming a small number of demonstrations to fine-tune each LLM to each domain?
2) What is the sensitivity of the performance of each LLM to chosen prompts? It seems like prompts used for evaluation are very complex and it would be good to understand how they were chosen and how prompts would effect the performance of each LLM.
3) Can the authors come up with a method to try tackle all of these benchmarks?
4) Can the authors explain more why the chosen set of text benchmarks comprehensively evaluate the ability of LLMs to act as agents? For instance, why isn't there a benchmark to test multiagent interaction between LLMs?

---

> ### Author Response · Authors · 2023-11-19
> **Response to Reviewer hDNu (1/4)**
>
> Dear Reviewer hDNu,
>
> Thank you for your comments and feedback. We appreciate your reviewing our paper and think our clarifications and improvements will address your concern:
>
> > Weakness 1:  The contributions of the paper seem very limited -- the paper **does not propose any new technical insights** and simply applies a variety of LLMs on many existing environments.
> >
>
> We are afraid that there might be some misunderstandings about our goal and contributions:
>
> 1. The comment ``simply applies a variety of LLMs on many existing environments'' isn't  accurate.
>     1. **Most environments are newly created in this work**: in AgentBench 5 out of 8 tested environments (OS, DB, KG, LTP, DCG) are created for the first time (see citation marks in Section 1), rather than ``many existing environments''.
>     2. **It is challenging to set up the evaluation**: to create testable new environments and integrate them for scalable testing on LLMs require quite arduous algorithmic and engineering efforts, rather than a ``simply'' achieved work.
> 2. Our contributions on technical insights are significant.
>     1. It is the very first work to quantitatively benchmark many LLMs (up to 27) on multiple real-world agent tasks in a unified setting.
>     2. The techniques we develop in creating 5 new environments out of 8 (Cf. Appendix B-F, 14 pages) are useful for later agent environment creation. The handy integrated testing toolkit we develop would serve as a cornerstone for all later new agent tasks to use.
>     3. The LLM behaviors and fall-shorts we discover and analyze during the evaluation are beneficial for improving LLMs’ agent abilities (Cf. Appendix J, 11 pages).
>
> > Weakness 2:  The **analysis is not particularly insightful** and I'm **not sure if the conclusions are fully accurate from the analysis**. For instance, task length exceeded may just be due to the fact that many LLMs are trained on short fixed context lengths. In setting such as this, it would be better to first summarize the long context to keep inputs in distribution to the input LLM.
> >
>
> Thank you for your feedback. We are afraid there might be some obfuscation about the `task length exceeded` you mentioned, which should refer to the term "Task Limit Exceeded (TLE)" in our paper as we understand. It has no relationship with LLMs' `short fixed context lengths` (i.e., the CLE as we reported, which only accounts less than 1% of all failure cases in most tasks as shown in Table 4), but holds the different meaning of "unexpected exceeding of maximum rounds of interaction as we design".
>
> In fact, we have carefully designed our tasks to be mostly solvable within around 1000 tokens, and a relatively harder limit is 3500 tokens when earlier histories begin to be folded. Our findings in Appendix J.2.3 proves that 95% of completed results are around this limit. Exceeding this limit typically indicates a divergence from the correct solution path (i.e., a failure solution).
>
> Notably, as shown in Figure 9 in Appendix J.2.4, in 90% of instances where Task Limit Exceeded occurred, the LLM began to deadly loop over previously generated content, resulting in repetition of identical contents in the final 10 rounds of interaction. It is the repetition that triggers the exceeding of maximum interaction rounds (i.e., TLE), rather than context length exceeding.
>
> We hope this clarification would address your concerns.

---

> ### Author Response · Authors · 2023-11-19
> **Response to Reviewer hDNu (2/4)**
>
> > Weakness 3:  Similarily, the fact that **code training helps models is relatively well known**. Codellamma performing worse than llama 2 may just be due to weakness in the original pretrained model that the model is fine-tuned on. Overall, **the conclusions in the paper mostly just follow the general trend of the more capable the LLM, the better the performance on the benchmark**
> >
>
> We are afraid there are two flaws in the comment’s description of our findings:
>
> 1. **CodeLlama is not worse than Llama2 in AgentBench**: Our findings indicate that CodeLlama generally surpasses Llama2 in performance. The exception is in agent tasks requiring deep thinking and language mastering (e.g., DCG), where coding training does harm to LLMs’ performance.
> 2. ********Our conclusion is not ``code training helps models''********: Exactly instead of following the general conclusion, we argue that code training has both *positive* and *negative* impacts on LLMs' performance on AgentBench (Cf. Section 4.3, as the DCG results show). In Section 1, we also emphasize that a ``**proper**'' strategy of introduce coding training is important.
>
> With regard to more novel insights, we provide them together with detailed analyses in Section 4.2 and Appendix J (11 pages), offering new perspectives on LLM capabilities and performance patterns. We summarize some novel findings as follow:
>
> 1. **Instruction Following Highly Affects LLM’s Ability as Agent**: We discover that most fail cases may due to the failure of instruction following. The easiest and quickest way for most LLMs to  boost their capability as agent is to carefully fine-tune on instruction dataset. (Cf. Appendix J.2.1)
> 2. **********The Gap Between Top and Best LLMs is Due to The Inconsistency of Planning and Acting:********** We qualitatively give, supplemented by case studies, differences in task planning and execution capabilities often exist between the best and the top LLMs. (Cf. Appendix J.2.2)
> 3. **Code Training has Complicated Impact**: We observe that code training significantly influences inferential generation and thought processes in models. For instance, models trained with code exhibit proficiency in tasks with static procedures but show limited performance in more dynamic tasks, suggesting a trade-off between procedural efficiency and general cognitive ability. (Cf. Section 4.3 & Appendix J.2.5)
> 4. **Unexpected Performance Parity in LLaMa-2 series**: We note the surprising similarity in performance between llama-2-13b and the much larger llama-2-70b. The reason behind might either due to an insufficient pretraining or a lack of enough instruction tuning of llama-2-70b. (Cf. Section 4.3**)**
> 5. **Less Capable LLMs Usually Fall into the Same Trap:** The majority of less potent LLMs typically end up engaging in repetitive, similar, or meaningless actions. (Cf. Appendix J.2.4)
> 6. **Completed Tasks Typically Require Only Limited Rounds and Tokens:** This implies that the majority of uncompleted tasks are due to a lack of requisite capabilities in Large Language Models (LLMs), rather than merely a deficiency in an adequate context window. (Cf. Appendix J.2.3)
>
> We also provide new perspectives and evidence for the following viewpoints:
>
> 1. **API-Based vs. OSS LLM Differences**: We analyze the performance disparities between API-based commercial LLMs and Open Source Software (OSS) LLM competitors, highlighting the distinct capabilities and limitations inherent in each type.
> 2. **Influence of High-Quality Alignment Data Training**: By comparing models like vicuna-13b and llama-2-13b, we conclude that training with high-quality alignment data, like that from ShareGPT, significantly enhances LLM performance. Vicuna-13b, aligned with ShareGPT data, notably outperforms its counterpart and rivals larger models. (Cf. Section 4.3)
>
> We wish our summarization and clarification would address your concern.

---

> ### Author Response · Authors · 2023-11-19
> **Response to Reviewer hDNu (3/4)**
>
> > Weakness 4:  In the related work section, it would be good to also add some references to **multiagent** approaches to reasoning such as multiagent debate. Also age 2023 seems to be incorrectly formulated.
> >
>
> Thanks for your great suggestion. We have updated related work and added multi-agent references in our revised version.
>
> > Weakness 5:  Some of the references in the paper are **incorrectly cited**. For instance, in the intro, **Reed et al. 2022** does not use a LLM to learn an embodied policy, a better reference would be [1]. Similarily, **Ahn et al 2022** does not use a complex multi-modal simulator based off games but rather a real robot.
> [1] Pre-Trained Language Models for Interactive Decision-Making. NeurIPS 2022
> >
>
> We appreciate your feedback on our paper. It seems there might be a misunderstanding about our context to cite these two papers. The section in our paper that references ``embodied agents'' is meant to encompass agents that interact with their environment through a physical presence within that environment, regardless of using LLMs. The articles by Reed et al. 2022 and Ahn et al. 2022 are discussed in this context of embodied agents. Furthermore, Ahn et al. 2022 specifically integrates language and vision modalities, exemplifying a multi-modal approach to embodied agents.
>
> Additionally, we are grateful for your recommendation of the paper “Pre-Trained Language Models for Interactive Decision-Making.“ We have included this reference in the revised version of our paper.
>
> > Question 1:  Have you tried seeing the results of each method assuming a small number of demonstrations to fine-tune each LLM to each domain?
> >
> >
> > Question 3:  Can the authors come up with a method to try tackle all of these benchmarks?
> >
>
> Our work does provide insights into the fine-tuning of models. For instance, in cases where tasks are not successfully completed, we observed interaction trajectories and identified two prevalent issues. Firstly, the models often fail to follow instructions, leading to outputs that either don't conform to the specified format or lack a valid action. Secondly, models tend to repeat previously given outputs after a certain number of turns, preventing the task from reaching a proper conclusion and causing interactions to hit the maximum turn limit. This suggests that the ability to follow instructions might be a critical aspect to focus on during the fine-tuning of LLM-as-Agents. Moreover, implementing penalties for repetitive content could be beneficial.
>
> However, due to our focus is to first set up a usable agent benchmark for the community, we would like to leave the fine-tuning exploration and the tackling of our benchmark in the future work. But it is noteworthy that some recent works, such as **FireAct[1]** and **AgentTuning[2]**, have been inspired and thus engaged in fine-tuning LLMs-as-Agents.
>
> > Question 2:  What is the **sensitivity of the performance of each LLM to chosen prompts**? It seems like prompts used for evaluation are very complex and it would be good to understand how they were chosen and how prompts would effect the performance of each LLM.
> >
>
> Our task prompts are chosen based on a combination of representative LLMs (rather than an individual series of models) to ensure their robustness. We experimented with various prompt paradigms, such as ReAct[3] and Chain of Thought (CoT)[4], and tested them across multiple models including claude-2, gpt-4, gpt-3.5-turbo, llama-2 and so on to ensure their universal validness. We did not select task prompts specifically tailored to any particular model. All models were tested under relatively equal conditions.

---

> ### Author Response · Authors · 2023-11-19
> **Response to Reviewer hDNu (4/4)**
>
> > Question 4:  Can the authors explain more **why the chosen set of text benchmarks comprehensively evaluate the ability of LLMs to act as agents?** For instance, why isn't there a benchmark to test **multiagent interaction** between LLMs?
> >
>
> Thank you for your question.
>
> - Task selection: We conducted extensive research and evaluated numerous tasks that could potentially form part of our benchmark, as we have heavily discussed in Section 1 & 5. Generally we follow two principles to make decisions:
>     - Recognition: We first consider task types that are universally acknowledged as crucial in the language agent community. For instance, NL2Bash[6] and InterCode[7] underscore the importance of OS tasks. Similarly, Wob[8], MiniWob++[9], Webshop[10], Mind2Web[11], and WebArena[12] highlight the critical role of web-based tasks.
>     - LLM Applicability: Another key criterion is that the task should be compatible and of proper difficulty levels for current LLMs’ few-shot text prompting with multi-round interactions. For example, though datasets like Minedojo[5] can provide more challenging open-world game environments, it generally requires vision information beyond existing LLMs capability.
> - Multi-agent evaluation: Thanks for your suggestion. Multi-agent interaction evaluation do count as an aspect in LLM-as-Agent evaluation, but we think currently it might be too hard for existing LLMs (especially those smaller ones with less than maybe 30B parameters, which cannot even handle single-agent tasks). Additionally, widely-acknowledged metrics and tasks for evaluating LLMs’ multi-agent are still not that clear at this time. Therefore, we decide to first place our focus on single-agent evaluation at present, but envisage our benchmark as something that will grow and expand over time to incorporate multi-agent interaction scenarios in the near future.
>
> ### References
>
> [1] Chen, Baian, et al. "FireAct: Toward Language Agent Fine-tuning." *arXiv preprint arXiv:2310.05915* (2023).
>
> [2] "AgentTuning: Enabling Generalized Agent Abilities for LLMs." *arXiv preprint arXiv:2310.12823* (2023).
>
> [3] Yao, Shunyu, et al. "ReAct: Synergizing Reasoning and Acting in Language Models." *The Eleventh International Conference on Learning Representations*. 2022.
>
> [4] Wei, Jason, et al. "Chain-of-thought prompting elicits reasoning in large language models." *Advances in Neural Information Processing Systems* 35 (2022): 24824-24837.
>
> [5] Fan, Linxi, et al. "Minedojo: Building open-ended embodied agents with internet-scale knowledge." *Advances in Neural Information Processing Systems* 35 (2022): 18343-18362.
>
> [6] Lin, Xi Victoria, et al. "NL2Bash: A Corpus and Semantic Parser for Natural Language Interface to the Linux Operating System." *Proceedings of the Eleventh International Conference on Language Resources and Evaluation (LREC 2018)*. 2018.
>
> [7] Yang, John, et al. "InterCode: Standardizing and Benchmarking Interactive Coding with Execution Feedback." *arXiv preprint arXiv:2306.14898* (2023).
>
> [8] Shi, Tianlin Tim, et al. "World of bits: an open-domain platform for web-based agents." *Proceedings of the 34th International Conference on Machine Learning-Volume 70*. 2017.
>
> [9] Liu, Evan Zheran, et al. "Reinforcement Learning on Web Interfaces using Workflow-Guided Exploration." *International Conference on Learning Representations*. 2018.
>
> [10] Yao, Shunyu, et al. "Webshop: Towards scalable real-world web interaction with grounded language agents." *Advances in Neural Information Processing Systems* 35 (2022): 20744-20757.
>
> [11] Deng, Xiang, et al. "Mind2Web: Towards a Generalist Agent for the Web." *arXiv preprint arXiv:2306.06070* (2023).
>
> [12] Zhou, Shuyan, et al. "WebArena: A Realistic Web Environment for Building Autonomous Agents." *NeurIPS 2023 Foundation Models for Decision Making Workshop*. 2023.

---

> ### Author Response · Authors · 2023-11-22
> **Thanks for your great review!**
>
> Dear Reviewer,
>
> Thank you very much again for your kind and valuable reviews! We hope our previous response addresses your questions, from which this work has benefited a lot. If possible, we would be deeply grateful if you could help share suggestions to further make AgentBench a stronger work!
>
> Thank you again for your encouraging comments!
>
> Authors of Submission7101

---

> ### Comment · Reviewer_hDNu · 2023-11-28
> **Rebuttal Response**
>
> Dear Authors,
>
> Thank you for your comments on the paper. The rebuttal helps clarify some of the questions I had about context length being extended as well as the domains studied. I have also read the reviews of the other reviewers. However, my main concerns about the paper remain and I do not think this paper meets the bar for being accepted to a conference and I am maintaining my score (happy to defend my score to other reviewers).
>
> The overall paper lacks novelty, consisting of prompting LLMs on a set of environments, with the results following exactly the ones one would expect (code models outperforming existing models and better language models performing poorer ones).  I'm also still confused about the exact prompts selected in the paper -- they authors say they did extensive tests to find the prompts, but there is no evidence in the paper about this (a robustness evaluation would be nice).
>
> Also, the introduction still seems factually incorrect. The GATO model (Reed et al.) does not use a LLM, rather it just trains an autoregressive model of actions. Furthermore, both Saycan and Inner Monologue do not use an simulators at all -- rather, the observations and provided as prompts to the model.

---

### Meta-Review · Area_Chair_Tsd6 · 2023-12-06

**Metareview:**

This paper presents AgentBench, a suite of benchmarks for evaluating large language models (LLMs) as Agents. The main contributions of this benchmark is the unified interface, the 8 agent tasks (consisting of both existing benchmarks such as WebShop and several newly designed ones), as well as the extensive evaluation of both closed and open source LLMs. The evaluation results are accompanied with some analyses.

The reviewers agree with the timeliness and significance of the results, and are mostly positive about the contributions. There are concerns that the conclusions may not be very surprising, the results may be sensitive to the specific prompts, and the setup is limited to prompting (without studies on fine-tuning), all of which I genuinely agree. However, at the current point where the agent capabilities of LLMs are generally well comprehended yet lacking a quantitative benchmark, I believe this work still offers a timely reference point and could have high impact. Therefore, I recommend acceptance.

**Justification For Why Not Higher Score:**

The evaluation results are arguably somewhat expected and in a limited scenario (prompting), and may not directly inspire new approaches.

**Justification For Why Not Lower Score:**

The benchmark is timely and could be useful for future work.

---

### Decision · Program_Chairs · 2024-01-16

Accept (poster)